# Cancer Patients’ Prehospital Emergency Care: Post Hoc Analysis from the French Prospective Multicenter Study EPICANCER

**DOI:** 10.3390/jcm10051145

**Published:** 2021-03-09

**Authors:** Olivier Peyrony, Jean-Paul Fontaine, Eloïse Trabattoni, Lionel Nakad, Sylvain Charreyre, Adrien Picaud, Juliane Bosc, Damien Viglino, Laurent Jacquin, Saïd Laribi, Laurent Pereira, Sylvain Thiriez, Anne-Laure Paquet, Alexandre Tanneau, Elie Azoulay, Sylvie Chevret

**Affiliations:** 1Emergency Department, Saint-Louis University Hospital, Assistance Publique-Hôpitaux de Paris, 75010 Paris, France; jean-paul.fontaine@aphp.fr; 2Emergency Department, Saint-Joseph Hospital, 75014 Paris, France; eloise.trabattoni@gmail.com; 3Emergency Department, Henri Mondor University Hospital, Assistance Publique-Hôpitaux de Paris, 94000 Créteil, France; lionel.nakad@aphp.fr; 4Emergency Department, SAMU de Lyon, Edouard Herriot University Hospital, 69622 Lyon, France; sylvain_charreyre@hotmail.com; 5University Claude Bernard Lyon 1, 69007 Lyon, France; 6Emergency Department, SAMU, SMUR. Le Mans Hospital, 72181 Le Mans, France; apicaud@ch-lemans.fr; 7Emergency Department, SMUR. Libourne and Sainte Foy la Grande Hospital, 33243 Libourne, France; juliane.bosc@hotmail.fr; 8Emergency Department, Grenoble-Alpes University Hospital, 38043 Grenoble, France; dviglino@chu-grenoble.fr; 9HP2 INSERM U 1042 University Grenoble-Alpes, 38043 Grenoble, France; 10Emergency Department, Hospices Civils de Lyon, Edouard Herriot University Hospital, 69622 Lyon, France; laurent.jacquin@chu-lyon.fr; 11Emergency Department, Tours University Hospital, 37000 Tours, France; s.laribi@chu-tours.fr; 12Emergency Department, Bichat University Hospital, Assistance Publique-Hôpitaux de Paris, 75018 Paris, France; laurent.pereira78@gmail.com; 13Emergency Department, SMUR, Victor Provo Hospital, Roubaix Hospital, 59100 Roubaix, France; sylvain.thiriez@ch-roubaix.fr; 14Emergency Department, la Pitié-Salpêtrière University Hospital, Assistance Publique-Hôpitaux de Paris, 75013 Paris, France; annelaure.paquet@gmail.com; 15Sorbonne-UPMC-Paris VI University, 75005 Paris, France; 16Emergency Department, SMUR of Lorient and Quimperlé, Bretagne Sud Hospital Group, 56322 Lorient, France; a.tanneau@ghbs.bzh; 17Intensive Care Unit, Saint-Louis University Hospital, Assistance Publique-Hôpitaux de Paris, 75010 Paris, France; elie.azoulay@aphp.fr; 18Centre of Research in Epidemiology and StatisticS (CRESS), INSERM, UMR 1153, Epidemiology and Clinical Statistics for Tumor, Respiratory, and Resuscitation Assessments (ECSTRRA) Team, University of Paris, 75006 Paris, France; sylvie.chevret@u-paris.fr; 19Department of Biostatistics and Medical Information, Saint-Louis University Hospital, Assistance Publique-Hôpitaux de Paris, 75010 Paris, France

**Keywords:** cancer, oncology, malignancy, emergency, prehospital

## Abstract

Background: Very little data are available concerning the prehospital emergency care of cancer patients. The objective of this study is to report the trajectories and outcomes of cancer patients attended by prehospital emergency services. Methods: This was an ancillary study from a three-day cross-sectional prospective multicenter study in France. Adult patients with cancer were included if they called the emergency medical dispatch center Service d’Aide Médicale Urgente (SAMU). The study was registered on ClinicalTrials.gov (NCT03393260, accessed on 8th January 2018). Results: During the study period, 1081 cancer patients called the SAMU. The three most frequent reasons were dyspnea (20.2%), neurological disorder (15.4%), and fatigue (13.1%). Among those patients, 949 (87.8%) were directed to the hospital, among which 802 (90.8%) were directed to an emergency department (ED) and 44 (5%) were transported directly to an intensive care unit (ICU). A mobile intensive care unit (MICU) was dispatched 213 (31.6%) times. The decision to dispatch an MICU seemed generally based on the patient’s reason for seeking emergency care and the presence of severity signs rather than on the malignancy or the patient general health status. Among the patients who were directed to the ED, 98 (16.1%) were deceased on day 30. Mortality was 15.4% for those patients directed to the ED but who were not admitted to the ICU in the next 7 days, 28.2% for those who were admitted to ICU in the next 7 days, and 56.1% for those patients transported by the MICU directly to the ICU. Conclusion: Cancer patients attending prehospital emergency care were most often directed to EDs. Patients who were directly transported to the ICU had a high mortality rate, raising the question of improving triage policies.

## 1. Introduction

A growing number of people live with cancer. Increased survival comes at the price of complications, which often require emergency care [1,2,3]. These complications may have multiple causes making the diagnostic work-up challenging and, in some cases, putting patients at the high risk of becoming critically ill [4]. Therefore, cancer has become a focus of interest for emergency research [5,6]. Epidemiological data have recently been published about cancer patients visiting emergency departments (EDs) [7,8,9,10,11,12], but less has been said about cancer patients attended to by prehospital emergency services [13,14,15]. For example, Wiese et al. showed the best out-of-hospital palliative medical care was given by prehospital emergency physicians who had significant expertise in palliative and emergency medical care [13]. In another recent study, Chen et al. pointed out that cancer patients frequently seek emergency care, particularly during the first year of the malignancy diagnosis. The authors have also shown that patients transported by emergency medical services (EMS) are more likely to be admitted to the hospital than those transported by personal vehicles [15]. However, these studies focused on palliative emergency care cancer patients or compared their characteristics and disposition by mode of arrival to the ED. This multicenter national prospective study aimed to describe cancer patients who call prehospital emergency services and report their outcomes depending on their trajectories.

## 2. Methods

### 2.1. Objectives

The objectives were to report the trajectories of cancer patients attended to by prehospital emergency services and to describe their chief complaints, characteristics, and outcomes.

### 2.2. Study Design, Settings, and Participants

This was an ancillary study from a three-day cross-sectional prospective study [12]. From Tuesday the 6th to Thursday the 8th of February 2018, the French emergency services of the Initiatives de Recherche aux Urgences (IRU) study group prospectively included all the consecutive cancer patients they attended to. As described previously [12], in France, patients can present to the ED through self-referral or after having called the dispatch center Services d’Aide Médicale Urgente (SAMU), where an emergency physician decides the appropriate level of response by sending the patient either paramedics (ambulance or fire department) or a mobile intensive care unit (MICU), staffed by an emergency physician, a nurse, and a paramedic, for prehospital medical assistance when a life-threatening condition is suspected. Medical advice can also be provided, or the patient can be referred to a general practitioner or to the ED.

Patients of 18 years or older with solid or hematologic malignancy were included regardless of their reasons for seeking emergency care. Only patients with cancer in remission for more than 5 years were excluded.

Three types of emergency services participated, and patients could be included at the SAMU level, at the MICU level, or at the ED level. Two hundred eighty-seven emergency services participated, among which 45 were SAMUs, 104 MICUs, and 138 EDs. They included more than 2000 patients during the study period. The present study focuses on patients who were included at the SAMU or at the MICU levels or those who were included at the ED level but were addressed by the SAMU.

### 2.3. Outcome Measures and Analysis

There was no intervention, and the data presented in the tables were collected prospectively by the attending emergency physician. For patients directed or transported to the hospital, any admission to the intensive care unit (ICU) during the first 30 days of hospital stay and status on day 30 (deceased, still hospitalized, or discharged home) were abstracted.

Descriptive statistics are reported. Continuous variables are presented as medians with their interquartile range (IQR), and categorical variables as number and percentages. We compared the characteristics of patients depending on whether an MICU was dispatched or not in a bivariate analysis. These comparisons used the Mann–Whitney test for continuous variables and the chi-square test for categorical variables. Results are presented with odds ratios (ORs) and their 95% confidence intervals (95% CI). All the *p*-values were two-sided, with values of 0.05 or less considered as statistically significant. The data were analyzed with R v3.5.0 software (the R Foundation for Statistical Computing, Vienna, Austria).

### 2.4. Study Registration and Ethical Approval

Patients were included after giving informed consent. The study was registered on ClinicalTrials.gov (NCT03393260, accessed on 8th January 2018) and approved by the Institutional Review Board of the French Speaking Society for Respiratory Medicine—Société de Pneumologie de Langue Française (number CEPRO 2017-038).

## 3. Results

### 3.1. General Characteristics, Trajectories, and Reasons for Seeking Emergency Care

During the study period, 1081 cancer patients called the SAMU medical dispatch center. Inclusions were made at the SAMU level for 531 patients, at the MICU level for 115 patients, and at the ED level for 435 patients. The prevalence of calls made by cancer patients to the SAMU was 0.6% [0.3–0.9%] (ranging from 0 to 2.9%) and represented 1% [0–7%] of the MICU interventions (ranging from 0 to 50%).

General characteristics are shown in Table 1. Median age of patients was 72 years, and 85.7% had solid malignancies.

The patients’ initial locations and final destinations are summarized in Figure 1. When cancer patients called the SAMU, paramedics were sent 744 (69.5%) times and they transported the patients to the ED in 90% of the cases. An MICU was dispatched 212 (19.8%) times, and transported patients in 161 (75.9%) cases, mostly to the ED (87%, 54%) and to the ICU (41%, 25.5%). At least 949 (87.8%) were directed to the hospital, among which 802 (90.8%) were directed to an ED and 44 (5%) were transported directly to an ICU. Other destinations (not ED or general ICU) included specific cardiologic ICUs in 20 cases and stroke units or neurosurgery in 5 cases.

For the 51 patients who were already hospitalized when the SAMU was called (to request transportation to another department), an MICU was dispatched 49 (96.1%) times and transported the patients to the ICU in 27 (53%) of the cases.

Patients were not transported to their referring cancer center in 353 (44.5%) cases (287 missing data). The SAMU emergency physician who responded to the call or who was in charge of the patient when the MICU was dispatched had no access to the oncologic medical file in 429 (72%) cases (485 missing data).

The reasons for cancer patients sought emergency care by calling the SAMU are summarized in Table 2. The three most frequent reasons were dyspnea (20.2%), neurological disorder (15.4%), and fatigue (13.1%). Thirty-one (2.9%) patients died on the scene before or after they were attended.

Table 3 shows the bivariate analysis comparing the patients’ characteristics depending on whether or not an MICU was dispatched. There were no differences in both groups concerning variables related to malignancy type or stage, or to patient general health status. Conversely, an MICU was dispatched more frequently for patients with cardiac arrest, dyspnea, a neurological disorder, and thoracic pain or for critically ill patients.

### 3.2. Care Delivered by MICUs

Table 4 shows the care delivered on site for the 115 patients who were included at the MICU level. Among those patients, 11 (9.6%) had no intervention, investigation, or treatment. The emergency physician had no access to the patient’s oncologic medical record in 69 (62.7%) cases (5 missing data). He or she considered that contacting with the referring oncologist was unnecessary for 78 (83.9%) patients and necessary for 15 (16.1%) patients (22 missing data). When contact was considered necessary, the oncologist was not accessible for 8 (53.3%) patients. Among the 49 critically ill patients, specific information on the patient’s resuscitation status was mentioned 10 (21.7%) times (3 missing data). When mentioned, palliative status was noted for 8 (80%) patients.

### 3.3. Patients’ Outcomes

Among the 802 patients who were directed to the ED, 44 (6%) were admitted to the ICU during hospital stay (63 missing data). This admission occurred during the first 7 days for 41 (93.2%) of them. Among those patients directed to the ED and later admitted to the ICU, 29 (70.7%) were transported by paramedics and 11 (26.8) by MICUs. On day 30, 98 (16.1%) patients were deceased (194 missing data). Among the 510 survivors on day 30, 386 (77.2%) were discharged home and 114 (22.8%) were still hospitalized (10 missing data).

Mortality varied from 15.4% for ED patients who were not admitted to the ICU in the next 7 days to 56.1% for patients directly admitted to the ICU (Figure 2).

## 4. Discussion

This study focusing on cancer patients attended by French prehospital emergency services showed that 8 patients out of 10 were finally addressed or transported to an ED, even when an MICU was dispatched, and less than 5% were transported directly to an ICU. The decision to dispatch an MICU seemed generally linked to the reason for seeking emergency care and to the presence of severity signs rather than to the malignancy stage or to the patient general health status. Mortality varied widely depending on patient trajectory.

We could not find any studies centered on prehospital emergency attendance for cancer patients besides those focusing on palliative emergency care [13,14].

The prevalence of calls made by cancer patients to the SAMU was relatively low, but this could be explained by the fact that some of those patients may not have declared their cancer over the phone. Furthermore, this prevalence refers to the number of daily calls, some of which were not systematically attended to by an emergency physician aware of the ongoing study. Thus, some of the cancer patients may have been missed. The prevalence of cancer patient attendance by MICUs was also low. In some cases, when the number of interventions per day was low, the prevalence could reach up to 50% (for example, if there were 2 MICU interventions, among which 1 concerned a cancer patient). The prevalence was lower than that of cancer patients in the French population, which is estimated at 5% [1]. However, this estimation accounts for all the cancer patients, including those in remission for more than 5 years. For comparison, Chen et al. found that cancer patients represented approximately 1.5% of all EMS transports in a study conducted in a large academic ED associated with an EMS system in Michigan, United States [15]. The prevalence of cancer patients in the ED is somewhat higher and has been estimated between 3% and 4% in the United States and in France [7,8,12].

The two most frequent reasons for cancer patients called the SAMU were dyspnea and neurological disorders. As observed in other studies, these reasons stood among the most frequent chief complaints for cancer patients admitted to EDs with fever and pain [7,9,15]. Pulmonary and neurological symptoms may have a wide number of causes such as infection, malignancy progression, or treatment toxicity. Moreover, the high number of patients referred to the hospital, hospitalized, or deceased on day 30, suggests that cancer patients attending prehospital emergency care may be complex and need a substantial burden of care. Thus, emergency physicians should be trained with a specific curriculum during their university course work focusing on acute care of cancer patients [16]. Some authors evaluated the benefit of embedding an oncologist in the ED with discordant results on patients’ admission rates [17,18]. We think that acute complications of cancer patients need to be dealt with an emergency physicians in the same way as any other acute complications of another chronic disease. That said, cancer patients’ prognoses are not only related to the acute condition but also to the malignancy stage and to the patient general health status [12,19,20]. For this reason, seeking the expertise of an oncologist or hematologist may be helpful to better decide the most appropriate orientation for critically ill patients [21].

Interestingly, we reported a 56.1% mortality rate for cancer patients admitted directly to an ICU. This rate is higher than those published recently in studies, showing an improvement in survival for cancer patients admitted to the ICU over the last decades [20,22,23]. It is likely that patients attended to by MICUs, and directly transported to the ICU are critically ill. Indeed, the French prehospital emergency system is based on early medical assessment by phone with the option to send an emergency physician able to begin advanced life support and intensive care procedures on site when a life-threatening condition is suspected. Then, those critically ill patients can be transported directly to an ICU. It is also possible that the selection of patients directly admitted to the ICU by MICUs was not optimal and that triage policies need to be improved to select cancer patients more likely to benefit from life-sustaining interventions [20]. This is also supported by the fact that out of 49 critically ill patients, 8 patients with a palliative status were transported directly to the ICU. Thus, not only the reason for seeking emergency care and severity should be taken into account when deciding whether or not to admit a patient to the ICU, but also cancer stage, treatment options, and, above all, patient general health status. For this reason, access to the oncologic file is mandatory in the emergency setting in order to avoid admitting patients with a “do not resuscitate” order to the ICU, which could be unethical. Also, Wiese et al. showed that experienced emergency physicians in palliative care send significantly fewer patients with advanced cancer, requesting prehospital care for palliative emergency situations, to the hospital and to the ICU [13].

Among patients directed to the ED, some were admitted to the ICU within the next 7 days raising the question of why they had not been admitted directly, especially for those transported by MICUs. We might wonder whether those patients could have benefited from a direct ICU transfer without passing through EDs. Actually, indirect or delayed ICU admission of critically ill cancer patients has shown to be associated with higher mortality [24,25,26]. In our study, patients admitted to the ICU after being transported to the ED had a lower mortality than those admitted directly. However, as we noted above, the patients admitted directly to the ICU by MICUs may have been more critically ill than those transported to the ED. Unfortunately, our data did not allow us to adjust the effect of delayed ICU admission on potential confounders such as patient clinical severity.

## 5. Limitations

Our study has several limitations. First, patients were included at three different levels (SAMU, MICU, and ED) and variables could differ from one setting to another. Second, the evaluation of patients’ severity was somewhat subjective as it was based on the emergency physician’s clinical judgement and not on a validated and reproducible score. However, clinical judgment may sometimes be more accurate than scores, and data were assessed prospectively, reducing the risk of bias. Third, to maximize physicians’ adherence to the study, we reduced the number of abstracted variables. Thus, some confounding factors were probably missing. That said, we took into account variables that are frequently associated with cancer patients’ outcomes such as performance status or underlying malignancy stage. Fourth, the outcome was abstracted only for hospitalized patients and not for the entire cohort. Fifth, some variables related to malignancy status had a higher rate of missing data, highlighting the lack of access to the patient oncologic record. For these reasons, we were unable to perform a predictive model to assess the association between patients’ characteristics when they were attended to by the SAMU and ICU admission or 30-day mortality. This analysis was conducted in our previous study focusing on cancer patients included at the ED level [12]. In addition, we could not determine the impact of direct ICU admission on patients’ outcomes after controlling for confounding factors such as severity.

## 6. Conclusions

In this study, cancer patients attending prehospital emergency care in France were most often directed to the hospital and particularly to EDs. Patients who were directly transported to the ICU had a high mortality rate, raising the question of improving triage policies.

## Figures and Tables

**Figure 1 jcm-10-01145-f001:**
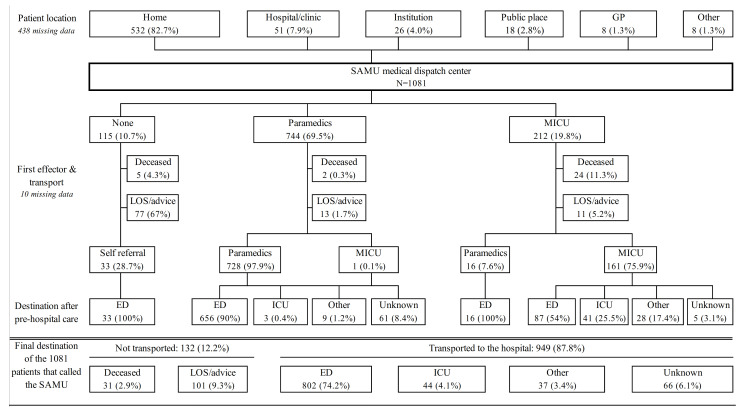
Flowchart of cancer patients who called the SAMU medical dispatch center. ED, emergency department; GP, general practitioner; ICU, intensive care unit; LOS, left on scene; MICU, mobile intensive care unit; SAMU, service d’aide médicale urgente.

**Figure 2 jcm-10-01145-f002:**
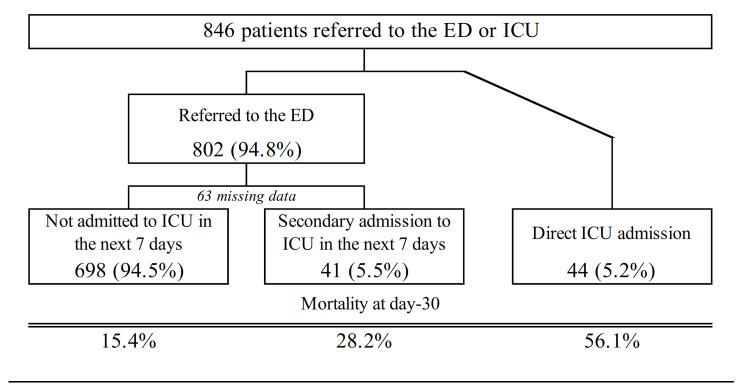
Flowchart of cancer patients’ mortality on day 30 depending on their trajectories.

**Table 1 jcm-10-01145-t001:** General characteristics of cancer patients who called the SAMU medical dispatch center.

			Missing Data
N	1081		
Age (years), Median (IQR)	72	(62–82)	23
Female Gender, *n* (%)	470	(43.8)	9
Night Shift (18 h–8 h), *n* (%)	412	(38.3)	5
Cancer Type, *n* (%)			13
Hematologic Malignancy	153	(14.3)	
Solid Malignancy	915	(85.7)	
Malignancy Status, *n* (%)			308
Complete or Partial Remission	417	(53.9)	
Uncontrolled	356	(46.1)	
Metastatic Malignancy, *n* (%)	301	(51.5)	497
Time since Malignancy Diagnosis, *n* (%)			198
<6 months	169	(19.1)	
6 months to 5 years	528	(59.8)	
>5 years	186	(21.1)	
Nursing Services, *n* (%)			92
None	552	(55.8)	
Home nursing service	338	(34.2)	
Nursing home care	42	(4.2)	
Institution	57	(5.8)	
Patient Alone at Home, *n* (%)	155	(27.0)	506
Poor Performance Status (>2), *n* (%)	234	(28.7)	267

IQR, interquartile range; SAMU, service d’aide médicale urgente.

**Table 2 jcm-10-01145-t002:** Reasons cancer patients called SAMU medical dispatch center.

			Missing Data
N	1081		
Reason, *n* (%)			2
Dyspnea	218	(20.2)	
Neurological Disorder	166	(15.4)	
Fatigue	141	(13.1)	
Trauma	129	(12.0)	
Gastro-intestinal	123	(11.4)	
Thoracic Pain	90	(8.3)	
Fever	75	(7.0)	
Bleeding	74	(6.9)	
Dizziness/Instability	74	(6.9)	
Pain	47	(4.4)	
Cardiac Arrest	27	(2.5)	
Agitation	21	(1.9)	
Metabolic Disorder	15	(1.4)	
Cytopenia	14	(1.3)	
Rash	12	(1.1)	
Shock	11	(1.0)	
Arrythmia	11	(1.0)	
Urologic Disorder	11	(1.0)	
Medical Device Complication	4	(0.4)	
Other	38	(3.5)	
Reason Related to Malignancy, *n* (%)	557	(54.7)	63

**Table 3 jcm-10-01145-t003:** Comparison of patients’ characteristics depending on whether or not an MICU was dispatched for patient evaluation and/or transport.

	MICU Dispatched				OR	95% CI		*p*	Missing Data
	No		Yes						
N	462		213						
Age (years), Median (IQR)	71	(62–82)	72	(64–80)	1.01	0.99	1.02	0.4	15
Male Gender, *n* (%)	248	(54.1)	129	(60.8)	1.32	0.95	1.84	0.1	5
Solid Malignancy, *n* (%)	399	(87.9)	173	(82.8)	0.66	0.42	1.05	0.08	12
Uncontrolled Malignancy, *n* (%)	142	(46.6)	72	(49.0)	1.10	0.74	1.63	0.6	223
Metastatic Malignancy, *n* (%)	125	(55.3)	54	(51.9)	0.87	0.55	1.39	0.6	345
Time since Malignancy Diagnosis, *n* (%)									161
<6 months	62	(18.6)	41	(22.8)	1.00				
6 months to 5 years	196	(58.7)	105	(58.3)	0.81	0.51	1.29	0.4	
>5 years	76	(22.8)	34	(18.9)	0.68	0.38	1.19	0.2	
Home Nursing Services, *n* (%)	183	(44.4)	75	(39.3)	0.81	0.57	1.15	0.2	72
Patient Alone at Home, *n* (%)	53	(24.0)	29	(28.4)	1.26	0.74	2.13	0.4	352
Poor Performance Status (>2), *n* (%)	86	(27.8)	48	(29.4)	1.08	0.71	1.64	0.7	203
Reason for Seeking Emergency Care, *n* (%)									2
Cardiac Arrest	7	(1.5)	20	(9.4)	6.71	2.92	17.33	<0.0001	
Dyspnea	70	(15.2)	74	(34.7)	2.97	2.03	4.34	<0.0001	
Neurological Disorder	57	(12.4)	47	(22.1)	2.00	1.3	3.06	0.001	
Thoracic Pain	37	(8.0)	32	(15.0)	2.02	1.22	3.35	0.006	
Trauma	57	(12.4)	4	(1.9)	0.14	0.04	0.33	0.0001	
Fatigue	58	(12.6)	5	(2.3)	0.17	0.06	0.38	0.0002	
Fever	38	(8.3)	5	(2.3)	0.27	0.09	0.63	0.006	
Digestive Disorder	60	(13.0)	10	(4.7)	0.33	0.16	0.63	0.002	
Reason Related to Malignancy, *n* (%)	248	(58.5)	115	(56.4)	0.92	0.65	1.29	0.6	47
Critically Ill, *n* (%)	29	(6.4)	115	(54.2)	17.42	11.1	28.1	<0.0001	8
Cardiac Arrest	7	(1.6)	21	(12.4)	8.64	3.77	22.3	<0.0001	73
Shock	1	(0.2)	41	(24.6)	138.62	29.7	>100	<0.0001	81
Respiratory Failure	12	(2.7)	65	(35.1)	19.23	10.41	38.48	<0.0001	52
Altered Mental Status	7	(1.6)	30	(17.2)	12.68	5.76	31.97	<0.0001	68

MICU, mobile intensive care unit; IQR, interquartile range; OR, Odds-Ratio; CI, Confidence Interval.

**Table 4 jcm-10-01145-t004:** Care delivered on site by the MICU.

			Missing Data
N	115		
Critically Ill, *n* (%)	49	(42.6)	0
Respiratory Failure	30	(26.1)	0
Shock	28	(24.6)	1
Altered Mental Status	14	(12.2)	0
Cardiac Arrest	13	(11.3)	0
Investigation, *n* (%)			0
Ultrasound	4	(3.5)	
Blood Sample	12	(10.4)	
ECG	51	(44.3)	
Venous Access, *n* (%)			0
Peripheral	78	(67.8)	
Central	4	(3.5)	
Long Term Central Catheter	2	(1.7)	
Intra-Osseous	1	(0.9)	
Oxygen Therapy, *n* (%)	56	(48.7)	0
Nasal	49	(42.6)	
Mechanical Ventilation	13	(11.3)	
Nasal High Flow Oxygen	8	(7.0)	
Existing Tracheostomy Use	1	(5.6)	
Treatment, *n* (%)			0
Fluid Challenge	24	(20.9)	
Catecholamines	13	(11.3)	
Analgesia (Not Morphine)	6	(5.2)	
Morphine	10	(8.7)	
Cardiopulmonary Resuscitation	8	(7.0)	
Sedation	7	(6.1)	
Antibiotics	2	(1.7)	
Other	11	(9.6)	
Length of Intervention (min), Median (IQR)	60	(42–82)	5

ECG: electrocardiogram.

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
