# Peer review of "Cancer Patients’ Prehospital Emergency Care: Post Hoc Analysis from the French Prospective Multicenter Study EPICANCER"

_jcm, 2021, doi:10.3390/jcm10051145_

Round 1

Reviewer 1 Report

The manuscript by Peyrony O. et al. focusing on cancer patients attended by prehospital emergency services shows that most patients were finally addressed or transported to an ED. It faces up a very interesting and important issue. The manuscript is clearly written and data are well collected. The discussion is reasonably supported by the results of the study.

Some specific concerns relating to the manuscript are detailed as following.

It would be interesting to investigate if some of the characteristics evaluated for the MICU dispatched are important also for the SAMU dispatched as for example for “patients alone at home” this could give useful information concerning the importance of the link between social problems and the request to the Emergency Department. So that I would be happy to see tab 3 replicate for SAMU dispatched patients. My personal feeling is that the main problem with patients affected by cancer is the waiting time in the Emergency Department without a real taking charge of specialised personnel, e.g. oncologist: Concerning this I recommend to the authors to consider a recent paper (ANTICANCER RESEARCH 38: 6387-6391 (2018)). The authors should discuss this issue.

Results, par.3.2, Last sentence: Concerning the MICU’s data what is more important, according me, is to discuss if it is deontological to transport to ICU 80% of patients in palliative status. This consideration is also supported by the mortality data. Please comment this data.

Discussion, 3rd paragraph, page 7. The following sentence is not clear “Thus, some of cancer patients may have been missed. The prevalence of cancer patient attendance  by MICUs was also low but could reach up to 50% in case of small number of interventions.”. Please, clarify this issue.

Discussion, 3rd paragraph, page 7: the comparison with the USA must be evaluated taking into account the different Health National System. Please, discuss this issue.

Discussion, 4rd paragraph, page 7: also concerning the sentence of this paragraph please consider the paper ANTICANCER RESEARCH 38: 6387-6391 (2018)).

The Limitation Section seems to me too significant. Not always acknowledging the limitations by simply numbering them release the authors, I suggest to explain better why the limitations do not weaken significantly the results of the paper.

Minor comments

Introduction, last line: focuses for “focuses”.

Fig.1 is not clear especially for the destination of the patients

Author Response

Comments and Suggestions for Authors

The manuscript by Peyrony O. et al. focusing on cancer patients attended by prehospital emergency services shows that most patients were finally addressed or transported to an ED. It faces up a very interesting and important issue. The manuscript is clearly written and data are well collected. The discussion is reasonably supported by the results of the study.

Some specific concerns relating to the manuscript are detailed as following.

> Thank you for your revision and constructing comments. We tried to respond to all of them and we hope that these changes will improve manuscript quality

It would be interesting to investigate if some of the characteristics evaluated for the MICU dispatched are important also for the SAMU dispatched as for example for “patients alone at home” this could give useful information concerning the importance of the link between social problems and the request to the Emergency Department. So that I would be happy to see tab 3 replicate for SAMU dispatched patients. My personal feeling is that the main problem with patients affected by cancer is the waiting time in the Emergency Department without a real taking charge of specialised personnel, e.g. oncologist: Concerning this I recommend to the authors to consider a recent paper (ANTICANCER RESEARCH 38: 6387-6391 (2018)). The authors should discuss this issue.

  • In the study, all the patients called the SAMU (dispatch center). Thus, it is not possible to compare the characteristics of the patients that called the SAMU from those who didn’t as we did in table 3 (comparing patients for whom a MICU was dispatched from those for whom a MICU was not dispatched after calling the SAMU). We agree that complications related to cancer may be overlooked in the ED and chief complaint attributed to easily to cancer progression. Also, other complications such as pain, anorexia, fatigue, digestive disorders may be neglected or overtreated. The paper of Legramante et al. is interesting but their results are discordant with another study (Brooks et al, Am J Emerg Med 2016;34:1934-1938) which found no difference in patient admission before or after embedding an oncologist in the ED. We think that emergency physicians should be trained with a specific curriculum during their university course focusing on emergency care of patients with cancer. Thus, cancer patients will be attended in the ED with the same rigorous bundle of care whatever the hour or day and without depending on the presence of an oncologist. Off course, oncologic advice is mandatory in some specific cases and close collaboration with onco-hematologists is necessary. But we think that acute complications of cancer patients need to be dealt by emergency physician such as acute complications of other chronic disease. However, we added this discussion in the 4th paragraph and cite these 2 papers.

Results, par.3.2, Last sentence: Concerning the MICU’s data what is more important, according me, is to discuss if it is deontological to transport to ICU 80% of patients in palliative status. This consideration is also supported by the mortality data. Please comment this data.

  > Those 80% palliative patients were 8 patients among 10 for whom patient’s resuscitation status was mentioned (that is 8 patients from the 49 critically ill, 16%). 39 critically ill patients had no indication concerning their resuscitation/palliative status. We agree that the high mortality rate in ICU is probably a reflection of the absence of triage policies when admitting cancer patients to ICU from pre-hospital care. We discussed this point in the 5th paragraph. According to your comment, we added a sentence pointing out this result and the ethical issue about admitting a patient with a DNR order to ICU.

Discussion, 3rd paragraph, page 7. The following sentence is not clear “Thus, some of cancer patients may have been missed. The prevalence of cancer patient attendance  by MICUs was also low but could reach up to 50% in case of small number of interventions.”. Please, clarify this issue.

  • We replace the sentence by: “In some cases, when the number of interventions per day was low, the prevalence could reach up to 50% (for example if the MICU did 2 interventions among which 1 concerned a cancer patient).”

Discussion, 3rd paragraph, page 7: the comparison with the USA must be evaluated taking into account the different Health National System. Please, discuss this issue.

  > The differences between both systems are detailed below in the 5th chapter.

Discussion, 4rd paragraph, page 7: also concerning the sentence of this paragraph please consider the paper ANTICANCER RESEARCH 38: 6387-6391 (2018)).

  • We added this reference and discussed it (and the paper of Brook et al with discordant results).

The Limitation Section seems to me too significant. Not always acknowledging the limitations by simply numbering them release the authors, I suggest to explain better why the limitations do not weaken significantly the results of the paper.

  > We developed this section as suggested.

Minor comments

Introduction, last line: focuses for “focuses”.

  • We changed it (focused instead of focussed in the last line of the objectives). We sent our manuscript for English editing (in progress).

Fig.1 is not clear especially for the destination of the patients

  • We modified the figure and clarified that destination was after pre-hospital care and that the lower part of the figure showed the final destination of all the patients that called the SAMU. We also added the number of transported/not transported patients with the %.

Reviewer 2 Report

It is a very interesting subject that often overlooked in the prehospital emergency services but getting more attention since cancer patients are increasing in numbers. Good topic to look into.

Author Response

Comments and Suggestions for Authors

It is a very interesting subject that often overlooked in the prehospital emergency services but getting more attention since cancer patients are increasing in numbers. Good topic to look into.

  • Thank you for your comments.

Reviewer 3 Report

Dear authors,

I read your manuscript with great interest. Improved triage tools in the out-of-hospital setting are urgently needed in my opinion, and I think your data contributes to that. There are a few issues that need to be addressed before further evaluating your work:

Overall

There are a few English grammar / style issues and some typos, please correct.

Introduction

-) What "diagnostic process" do you mean? Please clarify.

-) "... patients transported by EMS were more likely to be admitted to the hospital." More likely than who? Those left at home by the EMS crew?

Methods

-) You state that you excluded patients being in remission for over 5 years - why this number? Isn't this rather long? Can you sufficiently compare a patient being in remission for 4 years with an active form of disease?

Results

-) How is a cardiologic ICU different from a "normal" ICU in your country and why did you exclude it from the general ICU group?

-) Why would a MICU NOT be sent to a cardiac arrest in 7 cases or to a "critically ill" patient in 29 cases?

Discussion

-) You state that "even though" a MICU was dispatched, still a large percentage of patients was transported to an ED. You seem surprised - I think this needs some explanations what / what not EDs in your country provide. Do they to critical care / ALS / monitor and observe, etc.? Do they have ICU or IMCU capabilities?

-) You state that cancer patients were often brought to a hospital (and particularly to EDs) - is this surprising? What are your thoughts about this, also for future research?

Author Response

Comments and Suggestions for Authors

Dear authors,

I read your manuscript with great interest. Improved triage tools in the out-of-hospital setting are urgently needed in my opinion, and I think your data contributes to that. There are a few issues that need to be addressed before further evaluating your work:

> Thank you for your revision and constructing comments. We tried to respond to all of them and we hope that these changes will improve the quality of the manuscript

Overall

There are a few English grammar / style issues and some typos, please correct.

  • We sent our manuscript for English editing (in progress).

Introduction

-) What "diagnostic process" do you mean? Please clarify.

 > Diagnostic work-up is more appropriate. We changed it.

-) "... patients transported by EMS were more likely to be admitted to the hospital." More likely than who? Those left at home by the EMS crew?

 > There were more likely to be admitted than those transported by personal vehicle. We specified it.

Methods

-) You state that you excluded patients being in remission for over 5 years - why this number? Isn't this rather long? Can you sufficiently compare a patient being in remission for 4 years with an active form of disease?

> We agree that this cut-off is somewhat arbitrarily but we chose 5 years after discussion with our colleagues from oncology. Five years being the time period most often used for cancer survival and remission according to national cancer institute

Results

-) How is a cardiologic ICU different from a "normal" ICU in your country and why did you exclude it from the general ICU group?

 > In France, MICUs transport directly patients to the cardiologic or neurologic ICUs when they need an urgent procedure (e.g. cath-lab for ACS, thrombolysis for ischemic stroke) but not necessarily because they are critically ill. In our study, we wanted to focus on patients needing intensive care (shock, organ failure…). We clarify this point in the results section.

-) Why would a MICU NOT be sent to a cardiac arrest in 7 cases or to a "critically ill" patient in 29 cases?

 > We think that those patients with cardiac arrest maybe had a DNR order or were deceased when paramedics first arrived on scene, thus resuscitation was deemed futile by the SAMU that did not dispatch a MICU. For critically ill patient, it is more surprising, especially if palliative status or cancer status was not known. It is also possible that there was no available MICU and that SAMU dispatch paramedics to transport those patients quickly to the hospital. But we wanted to highlight with our study that cancer patients need the same pre-hospital urgent care when they are critically ill than other patients, especially when their underlying disease is controlled.

Discussion

-) You state that "even though" a MICU was dispatched, still a large percentage of patients was transported to an ED. You seem surprised - I think this needs some explanations what / what not EDs in your country provide. Do they to critical care / ALS / monitor and observe, etc.? Do they have ICU or IMCU capabilities?

 > Yes, in France, EDs may provide initial critical care and monitoring but, patients are rapidly admitted to ICU when they are intubated or need catecholamines.

-) You state that cancer patients were often brought to a hospital (and particularly to EDs) - is this surprising? What are your thoughts about this, also for future research?

 > It is surprising that the vast majority of cancer patients, that referred to a national dispatch center, was mostly conducted to an ED. One could have expected that those patients would have been orientated to a more specific location (oncology/hematology ward) or more often directly to ICU for critically ill patients attended by MICUs. Although, we observed that among patients directed to the ED and early admitted to the ICU, 1 in 4 patients were transported to the ED by MICU. It is difficult to infer that those patients could have been transported directly to the ICU because they may have become critically ill during hospital stay. But we think that ICU admission policies should be clarified early for those patients, since the pre-hospital setting and that they benefit from more specific orientations than EDs. For future research, it would be interesting to compare the pre-hospital cancer patients’ trajectories with non-cancer patients.